# Tofu Whey Wastewater as a Beneficial Supplement to Poultry Farming: Improving Production Performance and Protecting against *Salmonella* Infection

**DOI:** 10.3390/foods12010079

**Published:** 2022-12-23

**Authors:** Xuehuai Shen, Yayuan Xu, Lei Yin, Jianghua Cheng, Dongdong Yin, Ruihong Zhao, Yin Dai, Xiaomiao Hu, Hongyan Hou, Kun Qian, Xiaocheng Pan, Yongjie Liu

**Affiliations:** 1Institute of Animal Husbandry and Veterinary Science, Anhui Academy of Agricultural Sciences, Livestock and Poultry Epidemic Diseases Research Center of Anhui Province, Hefei 230001, China; 2College of Veterinary Medicine, Nanjing Agricultural University, Nanjing 210095, China; 3Anhui Province Key Laboratory of Livestock and Poultry Product Safety Engineering, Hefei 230001, China; 4Institute of Agro-Products Processing, Anhui Academy of Agricultural Sciences, Anhui Engineering Laboratory of Food Microbial Fermentation and Functional Application, Hefei 230001, China

**Keywords:** tofu whey wastewater, poultry, production performance, antioxidant capacity, immunity, gut microbiota, *Salmonella enteritidis* infection

## Abstract

Tofu whey wastewater (TWW) is a by-product of the tofu production process, and contains high amounts of organic products and *Lactobacillus ap*. However, no studies have been reported on whether naturally fermented TWW can be used as a beneficial additive for poultry production. This study analyzed the main nutritional components and microbial flora of naturally fermented TWW from rural tofu processing plants and their effect on chick production performance, role in modulating the biochemical and immune parameters, and protection against *Salmonella enteritidis* (*S. enteritidis*) infection. It was observed that the average pH of TWW was 4.08; therefore, the total viable count was 3.00 × 10^9^ CFU/mL and the abundance of *Lactobacillus* was 92.50%. Moreover, TWW supplementation increased the total weight gain and feed intake, reduced the feed/gain ratio, increased the length and relative weight of the gut, and reduced the colonization and excretion of *S. enteritidis* in chickens. Additionally, TWW decreased oxidative damage and pro-inflammatory cytokine secretion caused by *S. enteritidis* infection. In addition, TWW supplementation ensured the structure of the intestine remained relatively intact in *S. enteritidis*-infected chicken. Furthermore, TWW markedly promoted the intestinal barrier integrity and up-regulated the relative abundance of *Lactobacillus*, counteracting the changes in gut microbiota caused by *S. enteritidis* infection in chicken. In conclusion, our data demonstrated that TWW could be used as a beneficial addition to poultry production, providing a research basis for the further development of TWW as a health care application in in food-producing animal.

## 1. Introduction

Tofu is a traditional Chinese food. Tofu whey wastewater (TWW) is an abundant, nutrient-rich wastewater generated in the process of tofu production. TWW is regarded as an excellent alternative medium in fermentation, containing soluble sugars, proteins, fats, and inorganic salts [1]. However, due to its high organic concentration, chemical oxygen demand (COD) and biochemical oxygen demand (BOD), TWW may pollute the environment if not treated properly [2]. It has been estimated that 7–10 kg of TWW is produced per 1 kg of processed soybeans; thus, properly handling TWW is critical for soybean processing plants [3].

Recycling is an important measure used to solve the possible environmental pollution caused by TWW. Since TWW is a eutrophic substance, some studies have suggested using it as a medium for microalgae cultures to produce various products, and it has been proven to be safe and non-toxic [4]. Yu et al. used TWW as a medium for enhancing the vitamin B12 production and growth of *Propionibacterium freudenreichii* [5]. Moreover, Li et al. used a nitrogen source in TWW for high activity nattokinase production in *Bacillus subtilis* [6]. Wang et al. found that TWW is a better alternative medium for efficient algal culture compared with the regular green algae medium [7]. A mixture of saline wastewater (SWW) and tofu whey wastewater (TWW) presented better performance on DHA production as a medium for *Schizochytrium* sp. S31 cultivation [4]. However, due to the small production scale and scattered locations of tofu processing plants in many rural areas, the industrialization of TWW processing and utilization are still slow.

In the production process of tofu, adding fermented TWW to soybean milk is necessary. This fermented TWW contains many acid-producing microorganisms, and it is recycled in the tofu processing plant [8]. The recycling of fermented TWW ensures the consistency of the tofu processing and makes the types and proportions of microorganisms in the TWW produced by the processing plant relatively stable. However, in the tofu processing plant environment, the natural fermentation process of TWW is susceptible to nutrient content, storage conditions, temperature, and other organisms, which may cause significant changes in the main chemical composition and bacterial structure of TWW [9]. Fei et al. analyzed the microbial community in naturally fermented TWW with high-throughput sequencing technology and found that *Lactobacillus* is the predominant genus in the microbial community of naturally fermented tofu whey (NFTW), at 95.31%, while *Picha*, *Enterococcus*, *Bacillus*, and *Acetobacter* were present at values of 0.90%, 0.04%, 0.02%, and 0.09%, respectively [8]. *Lactobacillus_amylolyticus L6* isolated from TWW was confirmed to be highly viable in the human gastrointestinal tract and to have an inhibitory effect on pathogenic bacteria [10]. The use of *Lactobacillus* in animal production can improve production performance and gut health [11,12,13]. *Lactobacillus* secreted large amount of organic acid by consuming glucose, fructose, and most sucrose during TWW fermentation, of which lactic acid and acetic acid serve as the two main organic acids [8]. In addition, in the method of processing soybeans into tofu, soybean active peptides (SPs), a class of small protein fragments generated by in vitro enzymatic hydrolysis, fermentation, and food processing [14,15], may enter the TWW. For example, some researchers have extracted highly active lunasin from TWW [3], which have the capacity to prevent inflammatory-mediated disorders in the human gastrointestinal tract [16].

*Salmonella* is one of the most important zoonotic pathogens, which not only causes acute and chronic diseases in poultry flocks, but also transmits to humans from infected poultry [17]. Although different serovars have been associated with salmonellosis, a limited number are responsible for most human infections, among which *S. enteritidis* is one of the major serotypes of foodborne infections in the EU and USA [18,19]. Data from the China National Foodborne Diseases Surveillance Network showed that *S. enteritidis* has been the main serovar in nontyphoidal *Salmonella* infection in Zhejiang Province over the last 10 years [20]. Many studies have shown that *S. enteritidis* is widespread in poultry and can pose a threat to food and public health through the contamination of poultry meat and eggs [21,22,23]. Although naturally fermented TWW has been found to be rich in in nutrients (sugars, fats, proteins, etc.), organic acids and *Lactobacillus* sp., which may have beneficial effects on animal growth and health [8]. However, no research has been reported on whether the addition of TWW in poultry farming improves performance and reduces *S. enteritidis* infection, which poses a serious threat to poultry health and food safety in poultry flocks.

In this study, we analyzed the nutritional components and microbial flora structure of naturally fermented TWW from Huainan City (Anhui Province) rural tofu processing plants and its effects on chick production performance, organ index, antioxidant capacity, and immunity. We also evaluated the protective effect of TWW on *S. enteritidis* infection in chicks. The results of this study provide the basis for the cost-effective application of TWW in food animal production and health care, and also a new approach for the rational utilization of TWW.

## 2. Materials and Methods

### 2.1. Sample Preparation and Chemical Composition Analysis

Huainan City, Anhui Province, is the most famous tofu production region, also known as the “Hometown of Tofu”. There are many township tofu processing factories in Huainan, and the output of TWW is very large; thus, different TWW from Huainan were selected. Fresh TWW was collected from three township tofu processing factories in Huainan and was naturally fermented for 5 days at 25 °C. The samples were labeled D1 (Bagongshan), D2 (Shouxian), and D3 (Fengtai), respectively. The fermented TWW samples were collected, and their main chemical compositions, including PH, solid suspension, total sugar, reducing sugar, total nitrogen, total fat, lactic acid, and acetic acid, were analyzed. This work was completed at the Institute of Agricultural Products Processing, Anhui Academy of Agricultural Sciences.

### 2.2. Microbiota Analysis of TWW

Upon collection, fermented TWW samples were centrifuged (8000× *g* × 10 min) at 4 °C and stored at −80 °C. The DNA extraction and sequencing were performed by the Hangzhou Lianchuan Biotechnology Company. Briefly, DNA from different samples was extracted using the Hexadecyl trimethyl ammonium Bromide (CTAB) according to the manufacturer’s instructions (Thermo Fisher Scientific, Waltham, MA, USA). The total DNA was eluted in 50 μL of Elution buffer and stored at −80 °C until further measurement. PCR was performed with LC-Bio Technology Co., Ltd., (Hangzhou, China). The V3–V4 region of 16S rDNA was amplified using specific primers. The PCR products were purified (Beckman Coulter Genomics, Danvers, MA, USA) and quantified (Invitrogen, Waltham, MA, USA). The amplicon library was assessed on Agilent 2100 Bioanalyzer (Agilent, Santa Clara, CA, USA) and with the Library Quantification Kit for Illumina (Kapa Biosciences, Woburn, MA, USA). The sample libraries were sequenced using NovaSeq PE250 platform, and bioinformatics analyses were performed by LC-Bio (Hangzhou, China)

### 2.3. Chickens and Experiments Design

#### 2.3.1. Effects of TWW Supplementation on Chicken Performance, Organ Index, Antioxidant Capacity, and Immunity

One hundred and twenty 5-day-old specific-pathogen-free female chickens were purchased from Beijing Merial Vital Laboratory Animal Technology Co., Ltd. (China). Chickens were randomly divided into four groups —a control group (n = 30) and three TWW supplementation groups (n = 30/group)—which were given drinking water containing different concentrations of TWW (0, 5%, 10%, and 20% by volume). Each group contained 5 samples with 6 replicates each. The feed intake was recorded daily, and all animals were weighed on 0, 5, 15, and 20 d.

The feeding, management, inoculation, and euthanasia procedures of animal studies in this study were carried out in compliance with the regulations and guidelines of the Anhui Academy of Agricultural Sciences institutional animal care (AAAS2022-6), and strictly followed the requirements of the Laboratory Animal Guideline for ethical review of animal welfare (National Standards of the People’s Republic of China, GB/T35892-2018).

#### 2.3.2. Effects of TWW Supplementation on the Resistance of Chickens Infected with *S. enteritidis*

One hundred and twenty specific-pathogen-free 5-day-old female chickens were randomly divided into four groups: *S. enteritidis*, *S. enteritidis* + 10%TWW, 10%TWW, and control group. Each group contained 5 samples with 6 replicates each. The *S. enteritidis* and *S. enteritidis* + 10%TWW groups were orally administered 5 × 10^8^ CFU *S. enteritidis* (CVCC3377, China veterinary culture collection center). The final concentration 10%TWW was added to the drinking water in the *S. enteritidis* + 10%TWW and 10%TWW groups. No treatment was used for the control group. The experimental period was 7 days. The chickens were weighed and the feces samples in each group were collected every day; the incidence of the chicks was recorded. 

### 2.4. Determination of Organ Index 

On 20 d, 10 chickens were randomly selected from each group. Euthanasia was performed by intravenous injection of pentobarbital (150 mg/kg). Chickens were weighed (body weight), and internal organs (including heart, liver, spleen, stomach, small intestine, kidney, pancreas, and bursa of Fabricius and thymus) were collected and weighed using an electrical scale (quantitative analysis at 0.01 g level). The formula for organ index: organ index = (organ weight/body weight) × 100%.

### 2.5. Determination of Biochemical Indicators

The malondialdehyde (MDA) content, superoxide dismutase (SOD), glutathione peroxidase (GSH-Px), lactic dehydrogenase (LDH), alanine aminotransferase (ALT) and aspartate aminotransferase (AST) activities in serum or tissue homogenate samples were measured using commercial detection kits (Nanjing Jiancheng Bioengineering Institute, Nanjing, China) following the manufacturer’s instructions. The myeloperoxidase (MPO) activity (Wuhan Colorful Gene Biotechnology Co., Ltd., Wuhan, China) and protein concentration (Beyotime Biotechnology, Shanghai, China) were measured in jejunum tissue samples using commercial kits according to the instructions.

### 2.6. Determination of Immune Parameters

The content of IgY, complement C3, tumor necrosis factor-α (TNF-α), granulocyte-macrophage colony-stimulating factor (GM-CSF), interleukin-6 (IL-6) in serum samples, and secretory IgA (sIgA) in homogenate samples of jejunum were detected using ELISA kits (Wuhan ColorfulGene Biological Technology Co., Ltd., Wuhan, China) in accordance with the instructions.

### 2.7. Determination of S. enteritidis in Feces Samples

Ten fresh feces samples were collected from each group. Total DNA was extracted from 0.5 g of feces using the TIANamp kit (Tiangen, Beijing, China). The invA gene copy number of *S. enteritidis* was detected by qPCR (Appendix A). The qPCR was performed using the SYBR Green qPCR kit (GeneCopoeia, Guangzhou, China) with a total reaction volume of 20 μL. The standard curves, CT values and amplification efficiencies were analyzed using ABI 7500 instrument software.

### 2.8. Histological Analyses of Intestinal Tissue

The jejunal tissues from each group of chickens were fixed in 10% formaldehyde for 24 h, and then were embedded in paraffin. All tissues were sectioned to obtain 4 μm-thick section samples (Leica RM2135, Wetzlar, Germany) and stained with hematoxylin and eosin (HE, Zhuhai Beso Biotechnology Co., Ltd., Zhuhai, China). The specific procedure was described in a previous study [24]. The slides were observed under a microscope (100× magnification, Nikon 80i, Tokyo, Japan). The nucleus was stained blue, and the cytoplasm was stained red. Intestinal villus height was measured by CaseViever software (3DHISTECH SlideViewer 2.5, Budapest, Hungary).

### 2.9. Immunofluorescence of ZO-1 in Jejunum Tissues

The sections of jejunal tissue were rehydrated, the paraffin was removed, and the sections were washed with PBST (PBS plus 0.1% Tween-20). The sections were treated with 3% hydrogen peroxide and blocked with 10% BSA at room temperature for 2 h. The rabbit polyclonal antibody against ZO-1 was diluted 1:500 with PBST containing 1% BSA and incubated with the sections for 1 h at 37 °C. Following this, the sections were washed 3 times with PBST and incubated with goat anti-rabbit fluorescently labeled secondary antibodies for 1 h in a dark environment. The slides were stained with DAPI (4′,6-diamidino-2-phenylindole) and analyzed with a fluorescence microscope.

### 2.10. Quantitative Real-Time PCR Analysis

Total RNA was extracted from jejunal and spleen tissues using the TRIZOL kit (Takara, Beijing, China). In total, 2 μg of RNA samples was sampled with PrimeScript™ RT Master Mix (Takara, Beijing, China) for reverse transcription and cDNA synthesis. PCR was performed using SYBR Green PCR Master Mix. Reaction. Primers for inducible *nitric oxide synthase (iNOS*), *interleukin-6* (*IL-6*), *zone occludens-1* (*ZO-1*), *Occludin, Claudin-1* and *β-actin* (internal control) are shown in Table 1, and the PCR reaction system was 20 μL. All samples were analyzed using the ABI 7500 Real-time Detection System (Applied Biosystems, Waltham, MA, USA), and each sample was analyzed in triplicates. The mRNA expression of the target genes was analyzed using the 2^− ∆∆Ct^ method, following normalization with β-actin gene. The primers were synthesized by General Biotechnology Co., Ltd. (Hefei, China).

### 2.11. Gut Microbiota Analysis

Jejunal contents in each group were collected and kept at −80 °C. DNA from jejunal tissue was extracted using the CTAB according to the manufacturer’s instructions (Thermo Fisher Scientific, Waltham, MA, USA). The PCR amplification, sequencing, and bioinformatics analysis were performed by LC-Bio (Hangzhou, China). The operating procedure was the same as the method described in 2.2. The gut microbial sequencing data were deposited in GenBank under accession no. PRJNA873241.

### 2.12. Data Analysis and Statistics

Data are shown as mean values ± standard error of mean (SEM). Significant differences among the different groups were analyzed by one-way analysis of variance (ANOVA). *p* < 0.05 was considered to be statistically significant.

## 3. Results

### 3.1. The Chemical Parameters and Microbial Composition of TWW

Naturally, fermented TWW was collected from three different townships (Bagongshan, shouxian, and Fengtai) in Huainan, and the main chemical parameters determined are shown in Table 2. The average pH of the three TWW samples was 4.08, and the average solid suspension was about 3.13 g/L. The contents of total sugar and reducing sugar were 4.81 g/L and 1.52 g/L, respectively. Total nitrogen and total fat contents were 0.60 g/L and 0.74 g/L. The contents of lactic acid and acetic acid were 2.67 g/L and 1.87 g/L.

Next, a high-throughput sequencing analysis of the microbial community structure of naturally fermented TWW was performed. As shown in Figure 1, The OUT numbers of the fermented TWW samples in the three regions were 71, 81, and 82, respectively (Figure 1a). The Venn diagram showed that the number of common OUTs was 50, and the unique OUTs of each of D1, D2, and D3 were 6, 8, and 4, respectively (Figure 1b). The main microbial genus in the sample is *Lactobacillus*, and its average abundance was about 92.50%, followed by *Acetobacter* (3.75%), *Burkholderiaceae* (1.75%), and *Actinobacteria* (0.45%); other microbial genera only accounted for 1.55% of the total flora abundance (Table 1 and Figure 1c). The number of viable bacteria in fermented TWW determined by the plate counting method was 3.00 × 10^9^ CFU/mL. The analysis of the *Lactobacillus* genus in the sample revealed that it mainly included *Lactobacillus_amylolyticus*, *uncultured_bacterium_g_Lactobacillus*, *Lactobacillus_buchneri*, *Lactobacillus_casei* and *Lactobacillus_panis* (Figure 1d). The above data indicate that the fermented TWW contained a variety of chemical substances and abundant microorganisms.

### 3.2. Effects of TWW Supplementation on Chicken Performance, Organ Index, Antioxidant Capacity, and Immunity

#### 3.2.1. Effects of TWW Supplementation on Chicken Performance

The growth performance of chicks in each group is shown in Table 3. The total weight gain and average daily gain of each TWW supplementation group were higher than those of the control group, but only 10% of TWW groups reached a significant difference (*p* < 0.05). Additionally, the total feed intake of each TWW supplementation group was higher than that of the control group, and significant difference in 10% TWW group (*p* < 0.05), and the overall feed/gain ratio was lower than that of the control group (*p* > 0.05). The feed/gain ratio of the 10% TWW group was the lowest, i.e., 0.17 lower than that of the control group. The above results showed that TWW supplementation could improve the growth performance of chicks and reduce the feed/gain ratio during the brooding period.

#### 3.2.2. Effects of TWW Supplementation on Chicken Organ Index

The analysis of the chicken organ index is shown in Table 4. The heart index of the 10% and 20% TWW groups was higher than that of the control group, and the 10% TWW group reached a significant difference (*p* < 0.05). However, TWW supplementation had no significant effect on the liver index of the chicks in each group. Moreover, the three-dose TWW supplementation groups had a significantly higher kidney index than the control group (all *p* < 0.05). The index of the stomach (proventriculus and gizzard) was decreased in all TWW groups compared with the control group, of which the 10% TWW group reached a significant difference (*p* < 0.05). Compared with the control group, no statistically significant difference was observed in the pancreas index of the TWW supplementation groups. In addition, 10% and 20% TWW supplementation significantly increased the total intestinal index (*p* < 0.05); the cecal index in all TWW addition groups was significantly higher than that of the control group (*p* < 0.05). Additionally, three doses of TWW addition obviously increased the length of the small intestine and the cecum compared with the control group (*p* < 0.05) (Appendix A).

#### 3.2.3. Effects of TWW Supplementation on Antioxidant Capacity

Compared with the control group, 10% and 20% TWW supplementation significantly increased the concentration of SOD and GSH-Px in serum (*p* < 0.05 and *p* < 0.01, respectively; Figure 2a,b) and decreased serum MDA content (all *p* < 0.01) (Figure 2c). In addition, TWW decreased the content of LDH in serum, but only the 10% TWW group reached a significant difference vs. the control group (*p* < 0.05; Figure 2d).

#### 3.2.4. Effects of TWW Supplementation on Chicken Immunity

The chick immune organ index is shown in Figure 3a–c. The index of the spleen and bursa was slightly elevated in the TWW-treated chicks compared to the control group, but the difference was not significant (*p* > 0.05). The thymus index was significantly higher in the 10% TWW-supplemented group compared to the control group (*p* < 0.05), while there was no statistically significant difference in the other two TWW supplementation groups.

The results of serum cytokine determination showed no significant effect of different concentrations of TWW supplementation on the level of complement C3 in serum (*p* > 0.05; Figure 3d). GM-CSF levels were significantly lower in the 10% TWW supplementation group compared with the control group (*p* < 0.01; Figure 3e). Similarly, the IL-6 levels were reduced in different TWW groups, but only 10% TWW group was this statistically significant (*p* < 0.01; Figure 3f). Compared to the control group, TWW supplementation obviously increased the IgY content in serum (*p* < 0.01) and sIgA content in jejunum tissues (*p* < 0.05), except the 5%TWW group for the latter (*p* > 0.05; Figure 3g,h).

### 3.3. Effects of TWW Supplementation on Chicken Infected with S. enteritidis

#### 3.3.1. Effects of TWW Supplementation on the Pathological Changes of Chicks Infected with *S. enteritidis*

Chick weight was significantly lower in the *S. enteritidis*-infected group compared to the control group (*p* < 0.05); however, supplementation with 10% TWW was effective in reversing the weight loss of chicks caused by *S. enteritidis* infection (*p* < 0.05; Figure 4a). Next, *S. enteritidis* discharge in feces was quantitatively detected with qPCR (Figure 4b). The *S. enteritidis* copy number in feces reached 7.41 × 10^2^ at 1d PI in the *S. enteritidis* group, and then it quickly increased to the highest value (1.82 × 10^5^) at 4 days PI. The number then declined, decreasing to 0.44 × 10^2^ at 7 days PI. In the *S. enteritidis* + 10% TWW group, the *S. enteritidis* copy number was 2.95 × 10^2^ at 1d PI, and then rapidly decreased; the copy number was below the detection limit in some samples at 7 d PI.

MPO activity in the serum of chicks from the *S. enteritidis*-infected group was significantly higher than that of the control group (*p* < 0.01) while 10% TWW significantly decreased the MPO activity when compared to the *S. enteritidis* infected group (*p* < 0.01; Figure 4c). The spleen and liver indexes were significantly higher than those in the control group after *S. enteritidis* infection (*p* < 0.01), and the spleen and liver index significantly decreased in the *S. enteritidis* + 10% TWW group compared with the *S. enteritidis* group (*p* < 0.01; Figure 4d,e).

The histopathological analysis further showed intestine villi shortening and structural disruption in the *S. enteritidis*-alone treatment group, but 10% TWW maintained a relatively intact structure and significantly increased intestinal villus height when compared to the *S. enteritidis*-alone infection group (*p* < 0.01; Figure 4f,g).

#### 3.3.2. Effects of TWW Supplementation on the Oxidative Stress Index in *S. enteritidis*-Infected Chickens

MDA levels in serum and jejunal tissue were significantly enhanced in *S. enteritidis*-infected chicks compared with the control group (*p* < 0.01); 10% TWW supplementation significantly reduced MDA levels in *S. enteritidis*-infected chicks (*p* < 0.01; Figure 5a,b). SOD and LDH activities were significantly increased in serum of chickens infected with *S. enteritidis* alone (*p* < 0.01), and 10% TWW significantly decreased serum SOD and LDH activity (*p* < 0.01) in *S. enteritidis*-infected chickens (*p* < 0.01; Figure 5c,d). Similarly, the ALT and AST activity in serum and tissues obviously decreased in 10% TWW supplementation chickens when compared to *S. enteritidis*-alone infection group (*p* < 0.01 and *p* < 0.05, respectively; Figure 5e,f).

#### 3.3.3. Effects of TWW Supplementation on the Immune Function in *S. enteritidis*-Infected Chickens

Compared with the control group, the inflammation-related cytokines’ (TNF-α and IL-6) concentrations in serum were obviously enhanced in chickens after *S. enteritidis* infection (*p* < 0.01), while TNF-α and IL-6 levels obviously decreased in TWW- supplemented chickens when compared to those that were just infected with *S. enteritidis* (*p* < 0.05 or *p* < 0.01; Figure 6a,b). Similarly, the mRNA expression levels of *IL-6* and *iNOS* in jejunum were obviously up-regulated in *S. enteritidis*-alone infection group (*p* < 0.01; Figure 6d,e). Compared with the *S. enteritidis*-alone treatment, TWW significantly decreased the *IL-6* and *iNOS* mRNA levels in *S. enteritidis*-infected chickens (*p* < 0.01). Compared with the control chickens, the sIgA level in jejunum tissue obviously decreased in chickens after being infected with *S. enteritidis* (*p* < 0.01), and TWW obviously increased the concentration of sIgA in *S. enteritidis*-infected chickens (Figure 6c). 

#### 3.3.4. Effect of TWW Supplementation on the Epithelial Tight Junction Protein Expression of Jejunum

The IF analysis demonstrated that 10% TWW supplementation dramatically increased the ZO-1 staining intensity and prevented substantial loss when compared to *S. enteritidis*-alone infection (Figure 7a). In addition, the mRNA levels of *ZO-1*, *Occludin*, and *Claudin-1* were obviously decreased in chickens infected with *S. enteritidis* alone (*p* < 0.05), while 10% TWW supplementation significantly increased the *ZO-1*, *Occludin*, and *Claudin-1* mRNA when compared to the *S. enteritidis*-infected group (*p* < 0.01; Figure 7b,c).

#### 3.3.5. Effect of TWW Supplementation on the Gut Microbiota

The Venn diagram showed that the distinctive OTU numbers in control, *S. enteritidis*, *S. enteritidis* + 10% TWW, and 10%TWW group were 679, 487, 216, and 137, respectively, and 38 OTUs were present among four groups (Appendix A). The Simpson and Shannon index, which is shown in Appendix A, represents the α-diversity of gut microbiota in four groups. The results showed that 10% TWW supplementation significantly reduced the diversity compared to the control and *S. enteritidis*-infected chickens (*p* < 0.01), while *S. enteritidis* could perturb the stability of the microbiota formed by the addition of TWW, although the difference was not significant (*p* > 0.05). The principal component analysis (PCA) showed that the β-diversity of gut microbiota was obviously changed in chickens receiving TWW (Appendix A). The genus level of gut microbiota further showed that TWW supplementation significantly upregulated the relative abundance of *Lactobacillus* and *Burkholderia* in control and *S. enteritidis*-infected chickens (*p* < 0.01) (Figure 8a–e). Meanwhile, the relative abundance of *Ruminococcaceae* significantly decreased in chickens receiving TWW when compared to control and *S. enteritidis*-infected chickens (*p* < 0.01). Compared with the control group, the abundance of *Lactobacillus* and *Burkholderia* were significantly decreased in the *S. enteritidis* group, while the abundance of *Ruminococcaceae* was significantly increased. The TWW was able to completely offset the changes in gut microbiota caused by *S. enteritidis* infection.

The PICRUSt predicted the differences of gut microbial functional profiles among four treatment groups (Appendix A), which showed that some metabolic pathways changed due to TWW supplementation, such as nucleotide transport and metabolism, lipid metabolism, signaling molecules and interaction, and membrane transport compared to the control and *S. enteritidis*-infected groups.

## 4. Discussion

TWW is nutrient-rich wastewater produced in the production process of tofu. In China, there is a very high consumption of tofu, resulting in the production of 7–10 times the amount of TWW, which contains high levels of organic products that pollute the environment [3]. Studies have shown that TWW contains sugar, proteins, oligopeptides, and phytochemicals [1]. As a result, TWW is an ideal place for microorganisms to multiply. Previous studies have shown that lactic acid bacteria (LAB) are dominant microbes causing the fermentation of TWW [25]. In this study, we first analyzed the content of the main nutrients of naturally fermented TWW from tofu processing plants in three regions of Huainan, Anhui Province. The average pH was 4.08, and the total sugar, nitrogen, and fat were 4.81 g/L, 0.60 g/L, and 0.74 g/L, respectively. The contents of lactic acid and acetic acid were 2.67 g/L and 1.87 g/L, respectively. Fei et al. showed that fermentation stage of *Lactobacillus* in TWW plays a major role in the changes in their chemical composition, including an increase in acidity value and organic acid, the reduction of soybean oligosaccharides, and bioconversion of isoflavone [8]. In this study, we used microbial 16S rDNA sequencing to further analyze TWW and found differences in the diversity of microorganisms in TWW samples from different regions. The OUT number of TWW samples from Huainan was lower than that of other Lu’an and Xuancheng. Venn’s analysis further suggested that most of the microbial genus were shared in TWW from different sources, but there were also unique types of bacteria. The differences in microbial diversity in TWW may originate from regional differences, which may also be an important reason for the differences in the flavor of tofu produced in different regions [26,27].

Our results showed that the average abundance of *Lactobacillus* was 92.4% in TWW from three different regions. Further analysis of Lactobacillus species indicated the presence of *Lactobacillus_amylolyticus*, *uncultured_bacterium_g_Lactobacillus*, *Lactobacillus_buchneri*, *Lactobacillus_casei*, and *Lactobacillus_pani*. Similar data were reported in previous studies on TWW [8,28]. Many species of *Lactobacillus* can be used as probiotics for human health care or animal production. Fei et al. found that *Lactobacillus_amylolyticus L6* can colonize and maintain high viability in the human gastrointestinal tract and has a significant inhibitory effect on pathogenic bacteria; thus, it may potentially be used as a probiotic [29]. Meanwhile, *Lactobacillus* sp. has been suggested as an ideal probiotic candidate for improving the performance and control of pathogens in poultry, such as pathogenic *E. coli* and *Salmonella*, that can limit poultry disease and further transmission to humans [11,30]. Tabashsum et al. suggested using *Lactobacillus_casei* to decrease the colonization of *Campylobacter jejuni* and *S. enterica* in poultry gut [12]. The tested *Lactobacillus_casei* showed strong antagonistic activity against *Salmonella sp* in the environment and showed a high potential for use in poultry against salmonellosis [31]. Based on previous studies, we hypothesized that the addition of TWW has potential benefits for chicken growth performance and resistance to *Salmonella sp.* infection.

In our study, the total weight gain and total feed intake of TWW at supplementation groups at different dises were higher than those of the control group. Among them, 10% TWW significantly increased the total weight gain of chicks, and the overall feed/gain ratio decreased by 0.17. The feed/gain ratio is one of the core indicators representing the efficiency of poultry farming [32]. Our results suggested that the TWW supplementation can improve chick production efficiency and reduce feeding costs. However, due to our short experimental period, the long-term effect of TWW on the growth performance of chickens, especially broilers, requires further research.

The organ index is an important indicator that reflects the functional status of organs [33,34]. In this study, 10% TWW significantly increased the heart index of chicken, while kidney index was significantly higher in all TWW supplementation groups compared to controls. Yet, TWW had no significant effect on the liver index of the chicks. The relative weight of the intestine is closely related to intestinal development and absorption capacity [35]. The present results showed that the addition of TWW can significantly increase the relative weight and length of the intestine and cecum. In agreement with this changing of the intestine, increased relative weights of various gastrointestinal tract segments have been previously observed in broilers fed high amounts of fiber and corn-resistant starch [35,36]. Iji et al. demonstrated that intestinal weight changes might be mainly related to differences in cell size and the protein synthesis rate [37]. An increase in intestinal length shifts the process of digestive and absorptive from proximal into distal segments in the gut lumen, increasing the efficiency of nutrient absorption [38]. In this study, TWW enhanced the digestive and absorptive capacities of the chick gut, which may be an important reason for the lower feed/gain ratio.

*S. enteritidis* infection can cause growth retardation, pathological changes, and even death in chicks [39]. In addition, poultry that survives infection may serve as reservoirs of infection and cross-contamination [40,41]. Over the past 20 years, *S. enteritidis* has been the major serotype causing foodborne salmonellosis infections in humans, and contaminated hen’s eggs have been considered the most common vector of the infection [22,42,43]. Recent studies showed that *S. enteritidis* could also survive on the outer shell surface of the eggs. Moreover, egg internal contamination can originate from *Salmonella* penetrating the eggshell or directly contaminating the egg contents before oviposition due to infection of the reproductive organs [44]. Consistent with Nakphaichit’s findings [45], chick infection experiments in this study showed that *S. enteritidis* infection promoted chick body weight loss and increased the liver and spleen index, which suggested persistent infection in chicks. However, TWW supplementation effectively ameliorated the body weight loss, reduced the amount of *S. enteritidis* discharge in feces, and relieved pathological weight gain of the liver and spleen, which demonstrated that TWW supplementation was effective in attenuating the inflammation response in *S. enteritidis*-infected chicken.

In this study, we found that *S. enteritidis* infection leads to a significant increase in serum MPO, which can be reduced by TWW. Moreover, the histopathological analysis showed that intestine villi shortening and structural disruption, a direct index of enteritis, were obviously relieved in *S. enteritidis*-infected chicken receiving TWW. Consistent with these results, TWW supplementation significantly reduced the disruption of intestinal structures in chickens infected with *S. enteritidis*. To sum up, these results indicate that TWW could protect chicks against *S. enteritidis*. Importantly, TWW can reduce the colonization and excretion of *S. enteritidis* in the digestive tract, blocking horizontal transmission and environmental pollution.

The antioxidant capacity of serum reflects the health of the body. Low antioxidant capacity leads to the accumulation of many free radicals and damaged cell function [46,47]. In our study, TWW supplementation increased the serum levels of SOD and GSH-Px, two important antioxidant enzymes scavenging oxygen free radicals and protecting the functional integrity of cells [48,49]. In the present study, we found that the levels of MDA and LDH, two indicators that reflect cytotoxicity and tissue damage [50], were significantly enhanced in the serum and jejunum tissue of chickens after *S. enteritidis* infection, while MDA and LDH were reduced in chicks treated with TWW. This suggests that TWW had a protective effect on oxidative stress.

ALT and AST in serum represent the situation of liver damage and dysfunction [51]. Previous studies showed that *S. enteritidis* could colonize the liver and spleen after infection from the digestive tract and cause damage to the colonized organs [52]. In this study, *S. enteritidis* infection in chicks led to the enlargement and damage of the liver, and the corresponding serum AST and ALT levels were also significantly higher than those of uninfected chickens. On the contrary, the addition of 10% TWW significantly alleviated the liver damage of chicks caused by *S. enteritidis* infection, which is related to the fact that TWW can significantly inhibit the colonization of *S. enteritidis* in the intestine.

The status of innate immunity is influenced by a variety of factors, such as nutrition [53], stress [54], and gut microbes [55]. Recent studies found that probiotics can preserve intestinal homeostasis by modulating the immune response and inducing the development of T regulatory cells [56]. Our results showed that different doses of TWW had no significant effect on the immune organ index of chicks; yet the addition of 10% TWW significantly increased the relative weight of the thymus. Our results also showed that TWW supplementation reduced the concentrations of GM-CSF and IL-6 in the serum of chicks but had no effect on the content of complement C3; cytokines (C3, GM-CSF, and IL-6) are important markers of non-specific innate immunity [57]. In addition, TWW significantly increased serum IgY and jejunal tissue sIgA levels. Wang et al. found that *Lactobacillus casei* enhances intestinal mucosal immunity, regulates cytokine balance, enhances intestinal immune function, and effectively reduces intestinal inflammation [58]. The above results suggest that TWW can regulate non-specific innate immunity and intestinal mucosal immunity in chicks. Increased levels of *iNOS* expression can contribute to inflammatory disease [59]. The present study showed that *S. enteritidis* infection remarkably increased *iNOS* expression levels in jejunum and TNF-α and IL-6 levels in the serum, while TWW could significantly reverse the inflammatory response caused by *S. enteritidis* infection.

The integrity of the intestinal mucosal barrier and the structure of the intestinal microbiota can significantly influence the health of the gut [60]. ZO-1 (zonula occludens 1), Occludin and Claudins, as the three most important tight junction proteins, are important protein molecules that are responsible for the integrity of the intestinal mucosal barrier and determine intestinal permeability, and any abnormalities in these lead to increased intercellular permeability [61]. In this study, 10% TWW addition significantly increased the mRNA levels of ZO-1, Claudin-1 and Occludin in the intestine compared with chickens in the control group, and *S. enteritidis* infection resulted in a decrease in the expression levels of three tight junction proteins in the chick intestine, which was reversed by TWW addition. In addition, many studies have reported that *Salmonella* infection causes disorder in the gut microbiota, which plays a key role in maintaining intestinal function and permeability [62,63]. In this study, TWW supplementation significantly up-regulated the relative abundances of *Lactobacillus* and *Burkholderia* and downregulated the abundance of *Ruminococcaceae* in chickens. Consistent with these results, TWW could significantly affect the intestinal microflora structure of *S. enteritidis*-infected chickens, such as an abundance of *Lactobacillus* up-regulation. Many studies reported that supplementation with *Lactobacillus* modulates innate intestinal immunity. Khan et al. [63] suggested that in-feed supplementations of probiotics (e.g., *Lactobacillus, Bifidobacterium*, and *Bacillus*) could modulate the intestinal microbiota to improve animal performance and enhance resistance to infection by intestinal pathogens, such as *Salmonella* and *Campylobacter*. Similar results have been found in other studies, *Lactobacillus* shows promising potential for use as a preventive probiotic added directly to the diet for the control of the colonization of *Salmonella* spp. in poultry [64,65]. The mechanisms of action of probiotics are effectuated through the production of organic acids, activation of the host immune system, and production of antimicrobial agents [66]. Studies have confirmed that organic acid supplementation in poultry farming can improve production performance and reduce the risk of *Salmonella* infection [67]. Therefore, the abundance of organic acids in TWW may also be one of the important mechanisms for its beneficial effects. From the above data, we speculated that TWW supplementation could protect the integrity of the intestinal barrier by regulating the intestinal microbiota of chicks and upregulating the expression level of tight junction proteins, thereby alleviating the pathological changes caused by *S. enteritidis* infection. However, TWW contains a variety of nutrients (sugars, fats, proteins, etc.), organic acids (lactic acid, acetic acid), highly active *Lactobacillus*, and even contains active peptides derived from soybeans, many of which have been reported in promoting growth and health in chickens [68,69,70,71]. Therefore, the effect of TWW on growth-promoting and resistance to *S. enteritidis* infection in chicks may not be attributed to a single substance, but rather to the combined effect of the multiple substances it contains.

In conclusion, our data demonstrated the main chemical parameters and microbial composition of TWW from tofu processing plants in three regions of Huainan, Anhui Province. TWW supplementation improves chick performance, reduces the feed-to-weight ratio, improves the activity of antioxidant enzymes, and modulates immune function by decreasing pro-inflammatory cytokine secretion. In addition, TWW supplementation maintains intestinal barrier integrity by regulating the intestinal microbiota and upregulating the intestinal epithelial tight junction protein expression level, which further prevent *S. enteritidis* infections. This information furthers our understanding of TWW from different regions in Huainan and confirms the potential of TWW as a beneficial supplementation for poultry to improve production performance and prevent foodborne pathogenic microbial infections, providing a research basis for further low-cost development of the application of TWW in food-producing animal.

## Figures and Tables

**Figure 1 foods-12-00079-f001:**
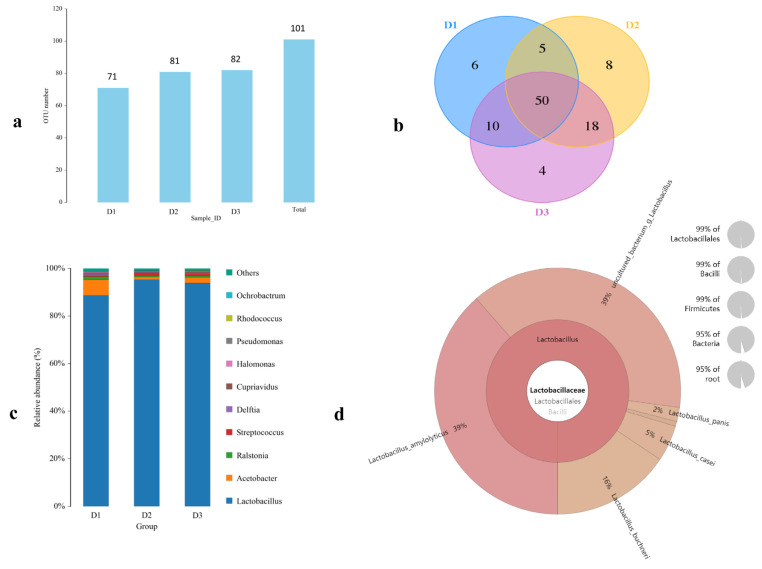
Analysis of microbial flora in naturally fermented TWW. (**a**) The OUT numbers of microbial flora. (**b**) Venn diagram analysis of microbial flora. (**c**) Relative abundance of microbial flora at the genus level. (**d**) Relative abundance of *Lactobacillus* species.

**Figure 2 foods-12-00079-f002:**
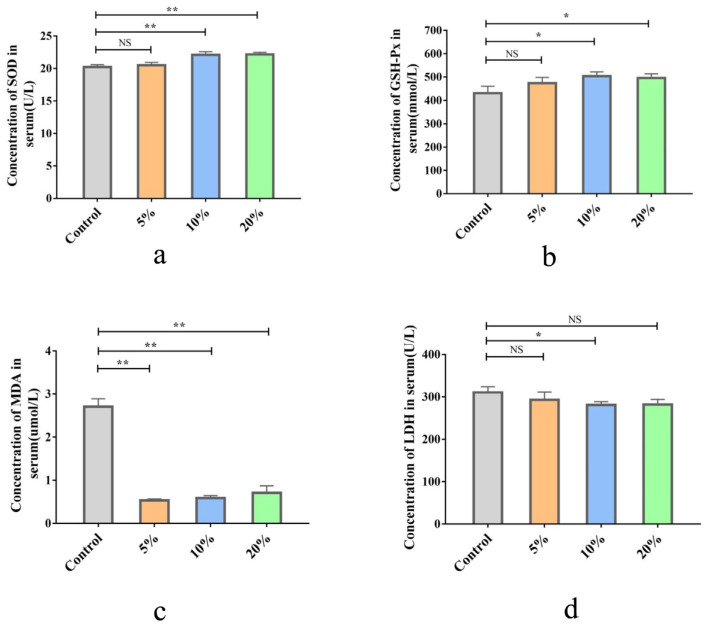
The effect of TWW on the antioxidant capacity in the serum of chickens. (**a**) Superoxide dismutase (SOD) activity. (**b**) Glutathione peroxidase (GSH-Px) activity. (**c**) Malondialdehyde (MDA) content. (**d**) Lactic dehydrogenase (LDH) activity. Values presented as means ± SEM. * *p* < 0.05, ** *p* < 0.01, compared with the control group. NS: no significance between the indicated groups.

**Figure 3 foods-12-00079-f003:**
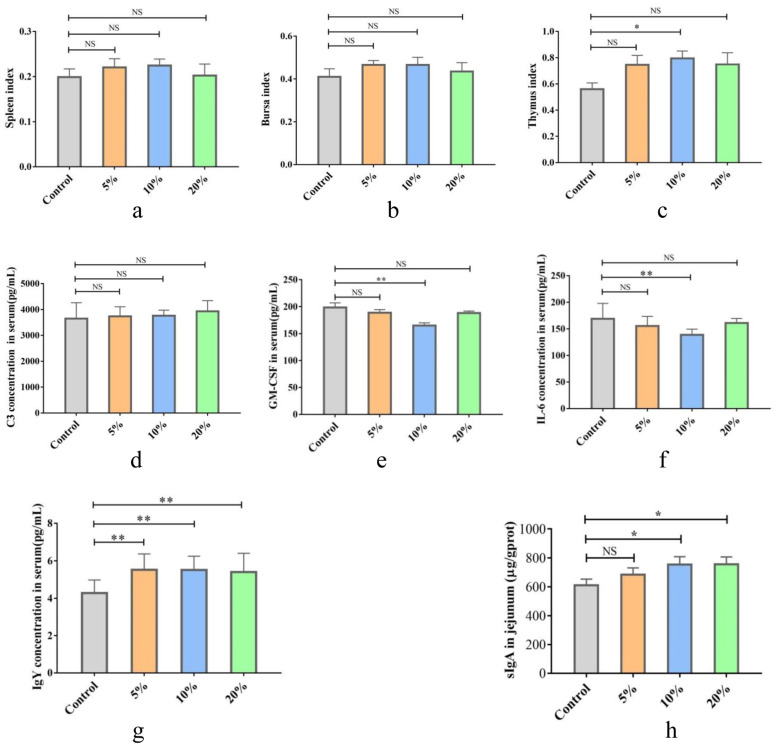
The effect of TWW on the immune indexes of chickens. (**a**–**c**) The relative weights of the spleen, bursa, and thymus, respectively. (**d**) Complement C3 content in serum. (**e**) Granulocyte-macrophage colony-stimulating factor (GM-CSF) content in serum. (**f**) Interleukin-6 (IL-6) content in serum. (**g**) Ig Y content in serum; (**h**) Secretory IgA (sIgA) content in jejunum tissues. Values presented as means ± SEM. * *p* < 0.05, ** *p* < 0.01, compared with the control group. NS: no significance between the indicated groups.

**Figure 4 foods-12-00079-f004:**
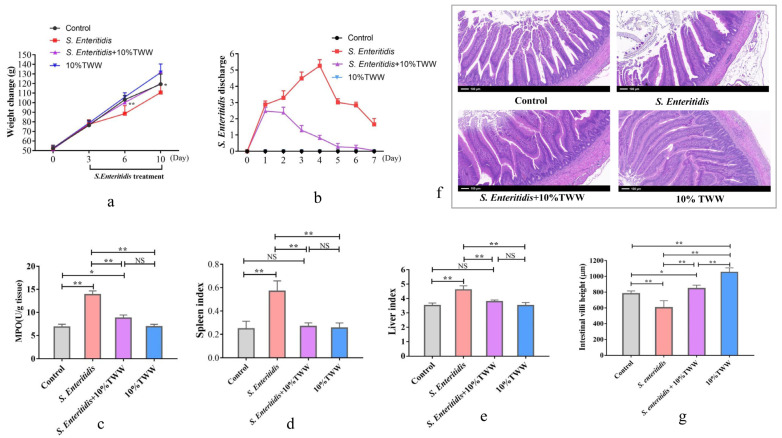
TWW supplementation reduces the pathological changes of chicks infected with *S. enteritidis*. (**a**) Body weight changes in chicken. (**b**) *S. enteritidis* discharge in feces (×log10cfu). (**c**) Myeloperoxidase (MPO) activity in serum. (**d**,**e**) The relative weights of the spleen and liver in chicken after *S. enteritidis* infection. (**f**) Histopathological examination of the jejunum (HE staining × 100). (**g**) The intestine villi height of jejunum. Values presented as means ± SEM. * *p* < 0.05, ** *p* < 0.01, compared between the indicated groups. NS: no significance between the indicated groups.

**Figure 5 foods-12-00079-f005:**
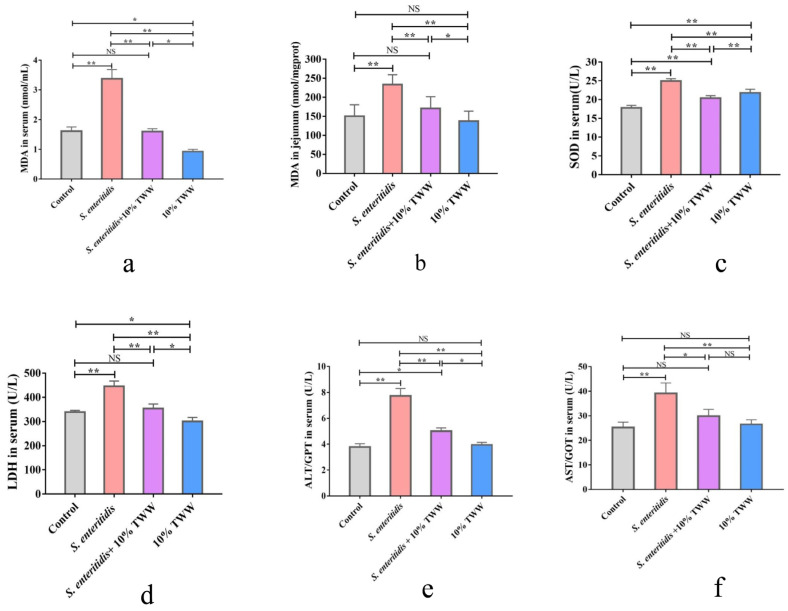
TWW ameliorates the oxidative stress index in *S. enteritidis*-infected chickens. (**a**,**b**) Malondialdehyde (MDA) content in serum and jejunum tissue. (**c**) Superoxide dismutase (SOD) activity in serum. (**d**) Lactic dehydrogenase activity in serum. (**e**,**f**) Alanine aminotransferase (ALT) and aspartate aminotransferase (AST) activity in serum. Values presented as means ± SEM. * *p* < 0.05, ** *p* < 0.01, compared between the indicated groups. NS: no significance between the indicated groups.

**Figure 6 foods-12-00079-f006:**
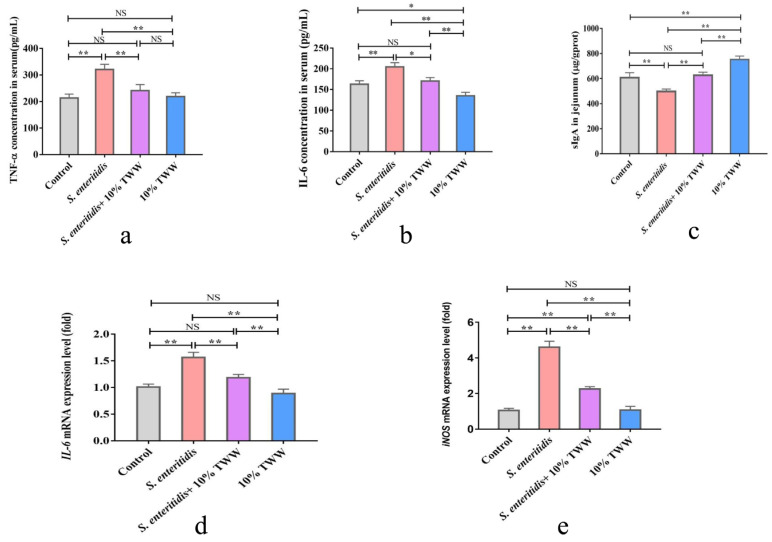
TWW improves the immune function in *S. enteritidis*-infected chickens. (**a**,**b**) Tumor necrosis factor-α (TNF-α) and interleukin-6 (IL-6) contents in serum. (**c**) Secretory IgA (sIgA) content in jejunum tissues. (**d**,**e**) The mRNA level of interleukin-6 (IL-6) and inducible nitric oxide synthase (iNOS). Values presented as means ± SEM. * *p* < 0.05, ** *p* < 0.01, compared between the indicated groups. NS: no significance between the indicated groups.

**Figure 7 foods-12-00079-f007:**
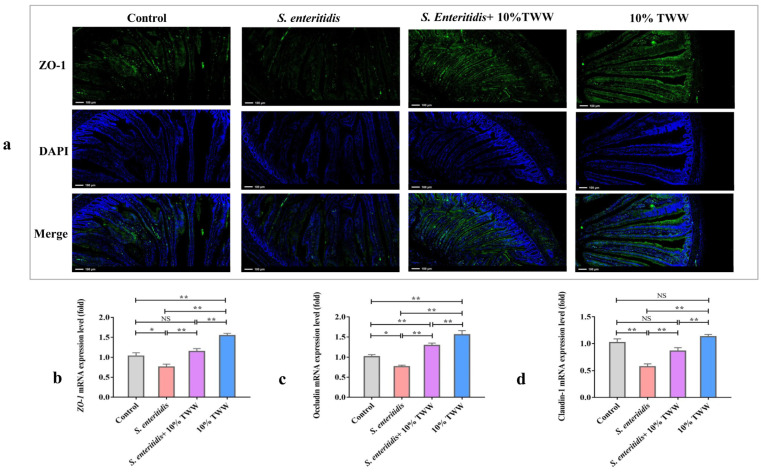
TWW enhances the tight junction protein expression level in the jejunum of *S. enteritidis*-infected chickens. (**a**) Sections of jejunum tissue were immunostained with anti-ZO-1 (red) and DAPI (blue) (100× magnification). (**b**–**d**) The mRNA levels of *ZO-1*, *Occludin*, and *Claudin-1* in the jejunum. Values presented as means ± SEM. * *p* < 0.05, ** *p* < 0.01, compared between the indicated groups. NS: no significance between the indicated groups.

**Figure 8 foods-12-00079-f008:**
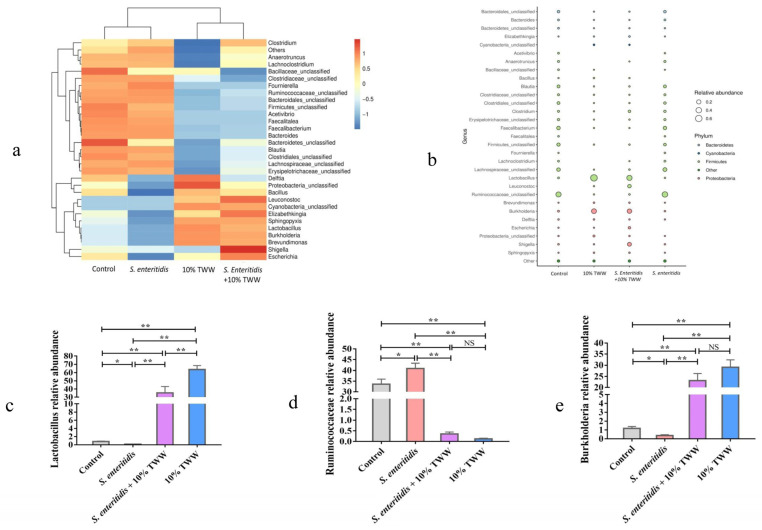
TWW alters the genus composition of gut microbiota in chickens. (**a**) Heatmap of a cluster of gut microbes in each group. (**b**) Bubble plot showing the relative abundance at the genus level. (**c**–**e**) Relative abundances of *Lactobacillus*, *Ruminococcaceae*, and *Burkholderia*. Values presented as means ± SEM. * *p* < 0.05, ** *p* < 0.01, compared between the indicated groups. NS: no significance between the indicated groups.

**Table 1 foods-12-00079-t001:** Primers used for real-time PCR analysis.

Gene Name b	Primer Sequence (5′to 3′)	Accession Number
*invA*	CCCGCTGCCGGTATTTGTTA	MK017942.1
	TCAGTCCTAACGACGACCCT	
*IL-6*	GATCCGGCAGATGGTGATAA	NM_204628.1
	AGGATGAGGTGCATGGTGAT	
*iNOS*	CCTGGGTTTCAGAAGTGGC	NM_204961.2
	CCTGGAGGTCCTGGAAGAGT	
*ZO-1*	CAGTCTGTGCGAAGTAGAGT	XM_046925209.1
	GAGGTTTGACAACTGACGTG	
*Claudin-1*	CGGAGTGAAACATCCTACCC	NM_001013611.2
	GGCATTGTAGTGTCCTCTCC	
*Occludin*	ACAAGCTCTTCCACATCAAG	NM_205128.1
	GGATGTGATTGAGTGTGTTT	
*β-actin*	ACGTCGCACTGGATTTCGAG	NM_205518.2
	TGTCAGCAATGCCAGGGTAC	

**Table 2 foods-12-00079-t002:** Chemical parameters and microbial composition of TWW.

Parameter	Value	Genus Ratio (%)	Value
PH	4.08 ± 0.26	Lactobacillus	92.50 ± 2.05
Solid suspension (g/L)	3.13 ± 0.46	Acetobacter	3.75 ± 1.36
Total sugar (g/L)	4.81 ± 0.36	Burkholderiaceae	1.75 ± 0.39
Reducing sugar (g/L)	1.52 ± 0.05	Actinobacteria	0.45 ± 0.10
Total nitrogen (g/L)	0.60 ± 0.02	Other	1.55 ± 0.61
Total fat (g/L)	0.74 ± 0.06	Total viable count (10^9^ CFU/Ml)	3.00 ± 0.28
Lactic acid (g/L)	2.67 ± 0.18		
Acetic acid (g/L)	1.87 ± 0.13		

Note: Values presented as means ± SEM.

**Table 3 foods-12-00079-t003:** The effect of FTWW on the growth performance of chicks.

Index	Groups			
	Control	5%	10%	20%
Total weight gain (g)	191.09 ± 9.78 a	199.69 ± 7.88 a	212.23 ± 2.51 b	197.19 ± 3.95 a
Average daily gain (g)	9.55 ± 0.49 a	9.98 ± 0.39 a	10.66 ± 0.13 b	9.86 ± 0.20 a
Total feed intake (g)	584.58 ± 5.42 a	595.35 ± 6.69 a	622.57 ± 2.52 b	594.98 ± 6.00 a
feed/gain (F/G)	3.09 ± 0.18 a	2.98 ± 0.10 a	2.92 ± 0.04 a	3.02 ± 0.07 a

Note: Values presented as means ± SEM. The presence of the same letter in a row represents no significant difference between groups (*p* > 0.05), and different letters in a row represent significant differences between groups (*p* < 0.05).

**Table 4 foods-12-00079-t004:** The effect of FTWW on the index of the metabolic and digestive organs of the chicks.

Organ Index	Groups			
	Control	5%	10%	20%
Heart	0.69 ± 0.05 a	0.69 ± 0.11 ab	0.78 ± 0.09 b	0.74 ± 0.05 ab
Liver	3.45 ± 0.42 a	3.61 ± 0.21 a	3.56 ± 0.49 a	3.37 ± 0.29 a
Kidney	1.14 ± 0.10 a	1.33 ± 0.19 b	1.30 ± 0.14 b	1.31 ± 0.14 b
Proventriculus and Gizzard	5.44 ± 1.06 a	5.26 ± 1.13 ab	4.37 ± 0.68 b	4.81 ± 1.19 ab
Pancreas	0.46 ± 0.08 a	0.47 ± 0.06 a	0.50 ± 0.07 a	0.54 ± 0.13 a
Total intestinal	6.70 ± 0.57 a	7.17 ± 0.42 a	7.26 ± 0.35 b	7.24 ± 0.69 b
Cecum	0.90 ± 0.07 a	1.43 ± 0.11 b	1.41 ± 0.04 b	1.42 ± 0.12 b

Note: Values presented as means ± SEM. The presence of the same letter in a row represents no significant difference between groups (*p* > 0.05), and different letters in a row represent significant differences between groups (*p* < 0.05).

## Data Availability

Data is contained within the article or Appendix A.

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
