# Peer review of "Tofu Whey Wastewater as a Beneficial Supplement to Poultry Farming: Improving Production Performance and Protecting against Salmonella Infection"

_foods, 2022, doi:10.3390/foods12010079_

Round 1

Reviewer 1 Report

Dear Editor,

The manuscript is well written with a lot of data and is highly relevant to the field. It uses waste water as a useful product for the improvement of production performance in poultry.

Some minor changes are suggested. 

1. Please elaborate clearly the Tables and figures. It is very difficult to read specifically figures because of small font size.

2. Please explain at the footnote of each Table about the present value(s).

Regards,

Reviewer 2 Report

The authors show the effect of using tofu whey wastewater supplementation in poultry farming, which indicates high productivity and immunity against Salmonella infection. The article is interesting and original. The method is well described in sufficient details. Moreover, the conclusion is really supported by the experimental data. However, there are some concerns about the writing and grammar in the attached file.

--- The authors show the effects of “Tofu”, Chinese food, as a dietary supplement to chicken for improving their productivity and protection against Salmonellosis.

As a general comment, the article sounds interesting contains novelty and the text is well-written. The method is not described in sufficient detail and is unclear. The conclusion needs to be improved according to the experimental data. There are some concerns that need to be clear for the redder as the following: -

-- Title, author, and affiliations:

--- Title should be shortened and to be more concise, in addition, Tofu has no direct role in the immunity of birds based on the current data, so the immunity concern should be omitted from the title. In addition, the title of the corresponding author should be omitted.

-- Introduction:

--- There is a very clear limitation of the article of using “Tofu supplement” is environmental pollution. So, the authors have to mention in the introduction these risks and, the trials used to provide this supplementation in a safe and healthy for the consumers; humans, and animals as well.

 --- The authors have to clarify very well, the mechanism (how) the Tofu food can modulate immunity in vivo although no significant difference in the data. If the author depended only on the similar action of probiotics, it does not make sense for Tofu, because the immune markers in the current study could not be evidenced for enhancing the immunity of birds.

--- The authors have to mention the exact chemical structures of Tofu, for example; sugars and fat, which kind of sugars and fat. Also, the authors should pay attention to the “Tofu”, because it is naturally available, so it might be affected by the temperature and humidity – The environment may accumulate fungi and in turn, will affect the quality and health. The authors have to explain in detail.

-- Material and Methods.

--- Authors have to mention the sex of chicks and used a bi-sex without differentiation, even for 5 days the authors can determine the sex of chicks because Tofu might affect the reproductive performance of chicks.

--- Since Tofu food can work in vivo as a probiotic supplementation for chickens, I suggest that authors compare probiotic group that is commercially available in China to see the difference in whether it might potentiate the efficiency of Tofu.

--- Regarding Statistical analyses, why the authors performed t-test among groups, It is assumed to be only ANOVA, and referring to that in the methods section.

--- Results

--- Why the authors weighted the internal organs like heart, kidney, while the title approaches the productivity which means the whole life and carcasses weights.

--- The authors have to mention carefully the purification of IgY and how they calculate the concentration.

Discussion and conclusion

--- The immune function of epithelial tight junction protein expression of jejunum is not clear; the authors have to mention the impact of this data.

--- It is well known that probiotics can block the intestinal villi receptors prior to the action of Salmonellosis and therefore protection the infection in advance. So, how the Tofu works?

--- The authors have to discuss the risks and the possibilities of preventive control of using Tofu for environmental and human sustainability.

--- I understand that the authors have limited data, but it would be great if they discuss (with reference) the role of Tofu in promoting behavior and birds ‘ activity for getting higher fitness and performance.

--- The author should omit reference # 56 because it is for mice not chickens, and they are totally different.

--- If the immune modulation of chicken is not clear after adding Tofu, so, the authors should not conclude that Tofu modulates the immunity of birds.

Reference section

--- The references should be updated until 2022

Tables

--- --- The authors have to revise the value ± SE or value ± SD in Table 2.

--- The authors have to revise the statistical differences with asterisk, P-values, in Table 3 and Table 4, because the values with no statistics.

--- The authors have to add a Table, that shows the costs and profits of using Tofu in chicken diet and put it in the main text of the article

Figures

--- Figure 1, Figure 8 have a bad resolution, they should be improved.

* In conclusion,making these concerns.
